# Characterization of neonatal opioid withdrawal syndrome in Arizona from 2010-2017

Emery R. Eaves[1,2,3☯], Jarrett Barber[3,4], Ryann Whealy[4,5], Sara A. Clancey[6], Rita Wright[7], Jill Hager Cocking[4,5], Joseph Spadafino[8], Crystal M. Hepp[3,4,5☯]*

1 Department of Anthropology, Northern Arizona University, Flagstaff, Arizona, United States of America, 2 Center for Health Equity Research, Northern Arizona University, Flagstaff, Arizona, United States of America, 3 Southwest Health Equity Research Collaborative, Northern Arizona University, Flagstaff, Arizona, United States of America, 4 School of Informatics, Computing, and Cyber Systems, Northern Arizona University, Flagstaff, Arizona, United States of America, 5 Pathogen and Microbiome Institute, Northern Arizona University, Flagstaff, Arizona, United States of America, 6 Institute for Human Development, Northern Arizona University, Flagstaff, Arizona, United States of America, 7 Department of Social Work, Northern Arizona University, Flagstaff, Arizona, United States of America, 8 Arizona Department of Health Services, Phoenix, Arizona, United States of America

☯ These authors contributed equally to this work.
* crystal.hepp@nau.edu

**Data Availability Statement:** Individual level data cannot be shared publicly because of identifiability concerns that could occur with individual level

## Abstract

In this paper, we describe a population of mothers who are opioid dependent at the time of giving birth and neonates exposed to opioids in utero who experience withdrawal following birth. While there have been studies of national trends in this population, there remains a gap in studies of regional trends. Using data from the Arizona Department of Health Services Hospital Discharge Database, this study aimed to characterize the population of neonates with neonatal opioid withdrawal syndrome (NOWS) and mothers who were opioid dependent at the time of giving birth, in Arizona. We analyzed approximately 1.2 million electronic medical records from the Arizona Department of Health Services Hospital Discharge Database to identify patterns and disparities across socioeconomic, ethnic, racial, and/or geographic groupings. In addition, we identified comorbid conditions that are differentially associated with NOWS in neonates or opioid dependence in mothers. Our analysis was designed to assess whether indicators such as race/ethnicity, insurance payer, marital status, and comorbidities are related to the use of opioids while pregnant. Our findings suggest that women and neonates who are non-Hispanic White and economically disadvantaged, tend be part of our populations of interest more frequently than expected. Additionally, women who are opioid dependent at the time of giving birth are unmarried more often than expected, and we suggest that marital status could be a proxy for support. Finally, we identified comorbidities associated with neonates who have NOWS and mothers who are opioid dependent not previously reported.

electronic medical record data. We have provided aggregate data in S1 Table. Hospital discharge data are housed at the Arizona Department of Health Services, and may be provided for future analyses upon approval of a Human Subjects Research Board protocol at the discretion of the Arizona Department of Health Services. Researchers wishing to reproduce or build on this study will need to submit a data request to the Arizona Department of Health Services to be approved: https://www.azdhs.gov/documents/director/administrative-counsel-rules/HSRB_NewProductSubmission.pdf.

**Funding:** This work was supported by a Pilot grant project to Crystal Hepp, Ph.D. as part of the NIH/NIMHD RCMI U54MD012388 (Julie Baldwin, Ph.D. PI). The authors acknowledge the contributions of Arizona Department of Health Services staff members Kyle Gardner and Timothy Flood for assistance defining the data request, the Northern Arizona University Information Technology Services for managing data security issues, and the Southwest Health Equity Research Collaborative Research Infrastructure Core for analytical assistance.

**Competing interests:** The authors have declared that no competing interests exist.

## Introduction

Neonatal Abstinence Syndrome (NOWS), also called Neonatal Opioid Withdrawal Syndrome (NOWS), is a consequence of abrupt withdrawal from intrauterine opioid exposure after birth [1–3]. Clinical abstinence symptoms are observed in 60–80% of substance exposed neonates, and include neurological, gastrointestinal, and autonomic complications [1]. Chronic opioid exposure in utero occurs in three different contexts: (1) active, untreated addiction to opioids (heroin or prescription opioids); (2) opioids used for chronic pain management; and (3) Medication Assisted Treatment (MAT) such as methadone or buprenorphine during pregnancy [4]. Common diagnostic criteria for NOWS include tremors, seizures, convulsions, feeding problems, vomiting, diarrhea, respiratory problems, and other neonatal complications [5]. In this paper, we describe the results of analysis of Arizona Hospital Discharge records from 2010 to 2017 to characterize the population of neonates born with NOWS and their mothers. Our findings suggest that other conditions may co-occur with NOWS more often than several of the commonly used diagnostic criteria. Our results suggest a need for better characterization of comorbid conditions in NOWS neonates and their mothers. Improving understanding of comorbid conditions and diagnostic criteria has implications for identification and secondary prevention of NOWS.

Nationwide, NOWS cases have been increasing exponentially and in national studies, the costs of NOWS births were at least three times greater than normal births [6]. NOWS births are more common in rural than in urban areas [7–10] and more likely to be among medicaid covered births [11]. Between 2004 and 2013, there was a 7-fold increase in NOWS in rural areas alone [12]. In Arizona, where 80% of the population live in mental health professional shortage areas [13], increasing availability of illicit drugs and steady rates of prescription opioid pain reliever use impact the population, including pregnant women [14]. The incidence of NOWS in Arizona continues to rise from 1.3 per 1,000 births in 1999 to 3.9 per 1,000 births in 2013, a threefold increase [15]. The number of opioid-related deaths in Arizona has increased 74% from 2012 to 2016, resulting in more than 2 deaths per day in 2016 [16]. Consistent with findings that socioeconomic status is a factor in NOWS cases [17], Hussaini and Saavedra reported that nearly 80% of NOWS cases in Arizona were paid for by Medicaid, especially in the border regions of the state [14].

On June 5, 2017, Arizona Governor Doug Ducey declared a Public Health State of Emergency due to the opioid epidemic [18]. An *Enhanced Surveillance Advisory* went into effect as a first step toward understanding the current opioid situation in Arizona and to collect data to develop best practices for interventions. As part of this advisory, any opioid-related event (opioid-related death, naloxone doses administered, NOWS cases, etc.) must be reported to Arizona Department of Health Services within 24 hours [18]. To inform the development of best practices in NOWS intervention in the context of increased attention, we conducted a comprehensive characterization of the population of neonates with NOWS and mothers who are opioid dependent. This analysis is the first of its kind in Arizona, and moves toward a better understanding of the population of Arizona neonates born with NOWS and their mothers. The analysis presented below is focused on the 5.5 years prior to and 1.5 years following the implementation of Arizona's opioid surveillance policy. While there have been studies of national trends in this population [17], regional trends and issues are less well understood [9, 19]. In this pilot project, we use data from the Arizona Department of Health Services Hospital Discharge Database to characterize the population of neonates born with NOWS and their mothers in Arizona from 2010 to 2017.

## Methods

### Data request

Northern Arizona University (NAU) has a data use agreement with the Arizona Department of Health Services (ADHS), allowing researchers an expedited path to access records in the ADHS Hospital Discharge Database and other databases. We submitted a data request to the Human Subjects Research Board (HSRB) at ADHS, to access electronic medical records for all neonates who were born and all mothers who gave birth in Arizona from 2010–2017. Notably, Indian Health Services hospitals are not required to report inpatient and emergency department visits to ADHS, so birth events at these hospitals are not captured. The request was approved as public health surveillance, and we additionally submitted a request for determination of non-human subjects research to the NAU Institutional Review Board. Based on the ADHS HSRB's determination of public health surveillance, NAU IRB determined the research to be non-human subjects research. This final dataset, which was transferred between two secure servers at ADHS and NAU, included the electronic medical records from 643,370 mothers and 663,353 neonates. All variables and descriptions included in the final dataset used in this project are included in S1 Table.

### Identifying the population of interest

The purpose of this study is to characterize the population of neonates with NOWS and mothers who are opioid dependent at the time of giving birth. We used insurance codes to identify subpopulations of interest within the larger mother and neonate dataset. The dataset spans 2010–2017, including both the International Classification of Diseases, Ninth Revision, Clinical Modification (ICD-9-CM, referred to as ICD9) and the International Classification of Diseases, Tenth Revision, Clinical Modification (ICD-10-CM, referred to as ICD10), as the change from ICD9 to ICD10 was required by all healthcare facilities in the United States no later than October 1, 2015. To identify neonates with NOWS, we used ICD9 and ICD10 codes 779.5 and P96.1, respectively. Similarly, to identify mothers who were opioid dependent at the time of giving birth, we used ICD9 codes 304.00–304.03 and 304.70–304.73 and ICD10 code F11, including all subcategories. These data include neonates of all gestational ages, including those in neonatal intensive care.

### Healthcare utilization

To better understand hospital resource utilization and to serve as a proxy for severity of morbidity, we compared length of stay and total charges of the subpopulations to the total population of Arizona mothers who have just given birth and neonates. The two variables were compared to each other using linear regression to better understand how well one explains the other, and t-tests were used to determine if the populations of interest had means that were significantly different.

### Demographic disparities

We conducted chi-square tests to determine if selected subpopulations belonged to specific racial and/or ethnic groups or used particular insurance payers significantly more often than expected. Similarly, we used a chi-square test to determine if mothers who were dependent on opioids at the time of giving birth had certain marital statuses more frequently than expected. Expected proportions were determined from the entire mother or neonate datasets (S1 Table).

## Geographic disparities

To identify geographic locations where there were more opioid dependent mothers at the time of giving birth than expected based on the total number of mothers who gave birth, we conducted a chi-square analysis. This analysis was completed for all non-tribal primary care areas in Arizona, aggregated from 2010–2017. A primary care area (PCA) is an area in which most residents seek primary health care from the same place. The Arizona Department of Health Services states that the PCA is meant to represent residents' "primary care seeking patterns" [20]. In addition, PCAs are aggregated to prevent re-identification of a patient in Arizona while allowing for resolution of population health issues at a scale better than that at which the geographically large Arizona counties provide.

## Associated comorbid conditions

In addition to demographic information, each inpatient and emergency department electronic medical record includes up to 26 ICD billing codes, including admitting and principal diagnosis codes, which could be any of approximately 13,000 ICD-9-CM or 69,000 ICD-10-CM codes. In the case of the selected subpopulations, these codes may include information regarding comorbidities of NOWS or opioid dependence. To understand comorbidity association with NOWS and opioid dependence, we selected comorbidities for their importance in classifying NOWS and opioid dependence as measured by their average minimum depth to the maximal sub-tree in classification random forests [21]. For this, we used the function var. select, with options method = 'md' and conservative = 'low', in the R library package random-ForestSRC [22, 23]. Our data present us with an imbalanced classification problem [24], wherein positive cases of NOWS or opioid dependence represent a minority of cases, with the majority of cases being negative. In such situations, overall classification performance—hence comorbidity selection—is dominated by the majority class, whereas our interest leans, instead, toward correct classification of the minority class. We use the method of balanced random forests [24], as implemented in the function imbalanced.rfsrc in the R library package random-ForestSRC [22, 23], to grow balanced classification random forests for NOWS and opioid dependence before computing the importance of comorbidities.

## Results

NOWS is increasing rapidly in Arizona. Opioid overdose in Arizona has been a cause for great concern, with suspected overdoses (n = 32,900, ~35 per day) and deaths (n = 3,935, ~4 per day) at epidemic levels from June 15, 2017 through January 16, 2020 [18]. The large number of neonates with NOWS born during the same period (n = 1,295), a consequence of the rise in opioid use, also warrants attention. To address this issue, we characterized the population of neonates with NOWS and mothers who were dependent on opioids at the time of giving birth within the context of the entire population of neonates born and mothers who gave birth in Arizona from 2010 through 2017.

### Healthcare utilization

To determine the impact of maternal opioid use during pregnancy on both neonatal and maternal morbidity as well as on healthcare utilization we compared hospitalization rates, average length of stay, and total charges of the entire populations versus the populations of interest. During the period of time represented in these data, the rate of newborn neonates who have NOWS has more than doubled from approximately 34 in 2010 to 88 in 2017, per every 10,000 births. (Fig 1). Similarly, the rate of mothers who are opioid dependent at the

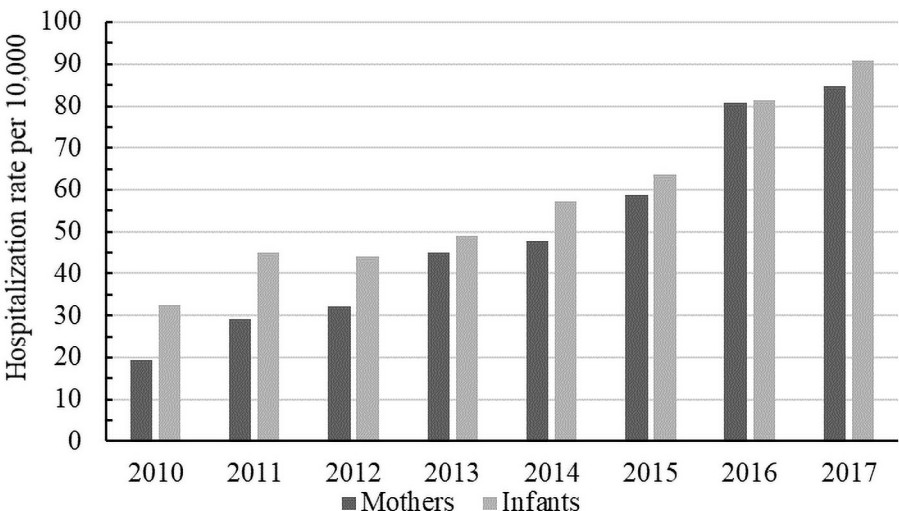

**Fig 1. NOWS hospitalization rates, per 10,000 births, in Arizona from 2010 to 2017.** Mothers who are dependent on opioids at the time of giving birth (dark grey) and newborn infants with NOWS (light grey).

time of giving birth has increased from 19 to 85 per every 10,000 mothers who have given birth (Fig 1). While reporting for neonates was substantially higher than reporting for mothers in 2010, hospitalization rates have evened out over time.

In Arizona, neonates with NOWS have an average length of stay (mean: 19.71 days, median: 16 days) approximately six times longer than that of all neonates (mean: 3.17 days, median: 2 days) (Table 1). Similarly, average total charges are also significantly higher for neonates who have NOWS (mean: $84,615, median: $49,887) in comparison to all neonates (mean: $10,784, median: $3,223). While neonates with NOWS only compose 0.5% of the total neonate population, the total charges associated with neonates who have NOWS account for 4.5% of all birth related charges ($323,230,298 of $7,153,221,072) from 2010 through 2017. Total charges and length of stay were also significantly higher for mothers who were dependent on opioids at the time of giving birth, however, the differences were modest in comparison to the neonate population.

## Demographic disparities

Previous studies characterizing neonates with NOWS and mothers who are dependent on opioids at the time of giving birth in the United States found that these populations were, more often than expected, non-Hispanic white and insured by Medicaid [25, 26]. We additionally examined these demographics to determine if the most heavily impacted populations in Arizona followed national trends. In agreement with previous studies, we found that both neonate

**Table 1. Comparison of hospital utilization variables in the target versus non-target populations and results of the t-test analyses.**

| POPULATION | LENGTH OF STAY (DAYS) | | | TOTAL CHARGES ($) | | |
|---|---|---|---|---|---|---|
| | Mean | Median | p-value | Mean | Median | p-value |
| INFANTS WITHOUT NAS | 3.07 | 2 | | 10356 | 3209 | |
| INFANTS WITH NAS | 19.71 | 16 | P<0.0001 | 84615 | 49887 | P<0.0001 |
| NON-OPIOID DEPENDENT MOTHERS | 2.44 | 2 | | 16801 | 14174 | |
| OPIOID DEPENDENT MOTHERS | 3.22 | 3 | P<0.0001 | 23674 | 18406 | P<0.0001 |

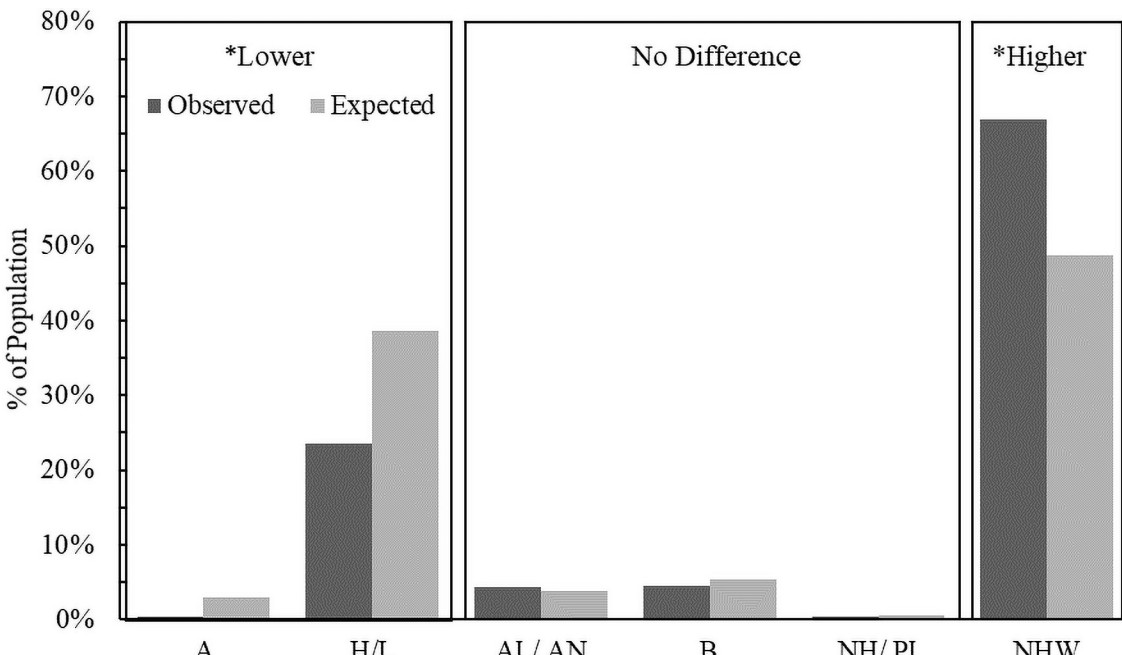

**Fig 2. Comparison of the observed versus expected proportions of NOWS in each racial/ethnic group of infants.** Boxes labelled higher or lower indicate that observed proportions are significantly higher or lower than expected proportions based on a chi-square analysis after post-hoc comparisons that incorporate a Bonferroni correction with six groups (p<0.001/6). A: Asian, H/L: Hispanic or Latino, AI/AN: American Indian or Alaskan Native, B: Black, NH/PI: Native Hawaiian or Pacific Islander, NHW: Non-Hispanic White.

(Fig 2) and mother (Fig 3) populations of interest were, significantly more often than expected, non-Hispanic white, and were Asian or Hispanic/Latino significantly less often than expected.

Our target populations were insured by either Medicaid or Medicare significantly more often and by private or military (TRICARE) insurance less often than expected (Figs 4 and 5).

We additionally considered maternal marital status and found that women dependent on opioids at the time of giving birth were unmarried significantly more often than expected based on the total population proportions while unmarried women were dependent on opioids significantly less often than expected (Fig 6). We suppressed data from categories where there were less than 10 mothers (i.e. widowed).

## Geographic disparities

Arizona is the 6<sup>th</sup> largest state in the US by area, but is composed of only 15 counties, where six are among the top 20 geographically largest counties in the United States. The result of a large state being spread into relatively few counties is that distinct human populations are forced into county level estimates which are unlikely to provide a relevant picture of population health. The ADHS has approached this issue by aggregating and reporting population health results for many conditions at the level of PCA. In an effort to identify if and where maternal opioid dependence is clustered, we adopted the ADHS strategy and compared counts across the 126 PCAs that compose Arizona. Within the entire maternal dataset, 25,936 records did not include a PCA, including 75 mothers who were dependent on opioids at the time of giving birth, and these records were not included in the geographic analysis. In addition, we suppressed statistically significant results for PCAs where there were fewer than 10 mothers who were dependent on opioids at the time of giving birth as well as those that are primarily

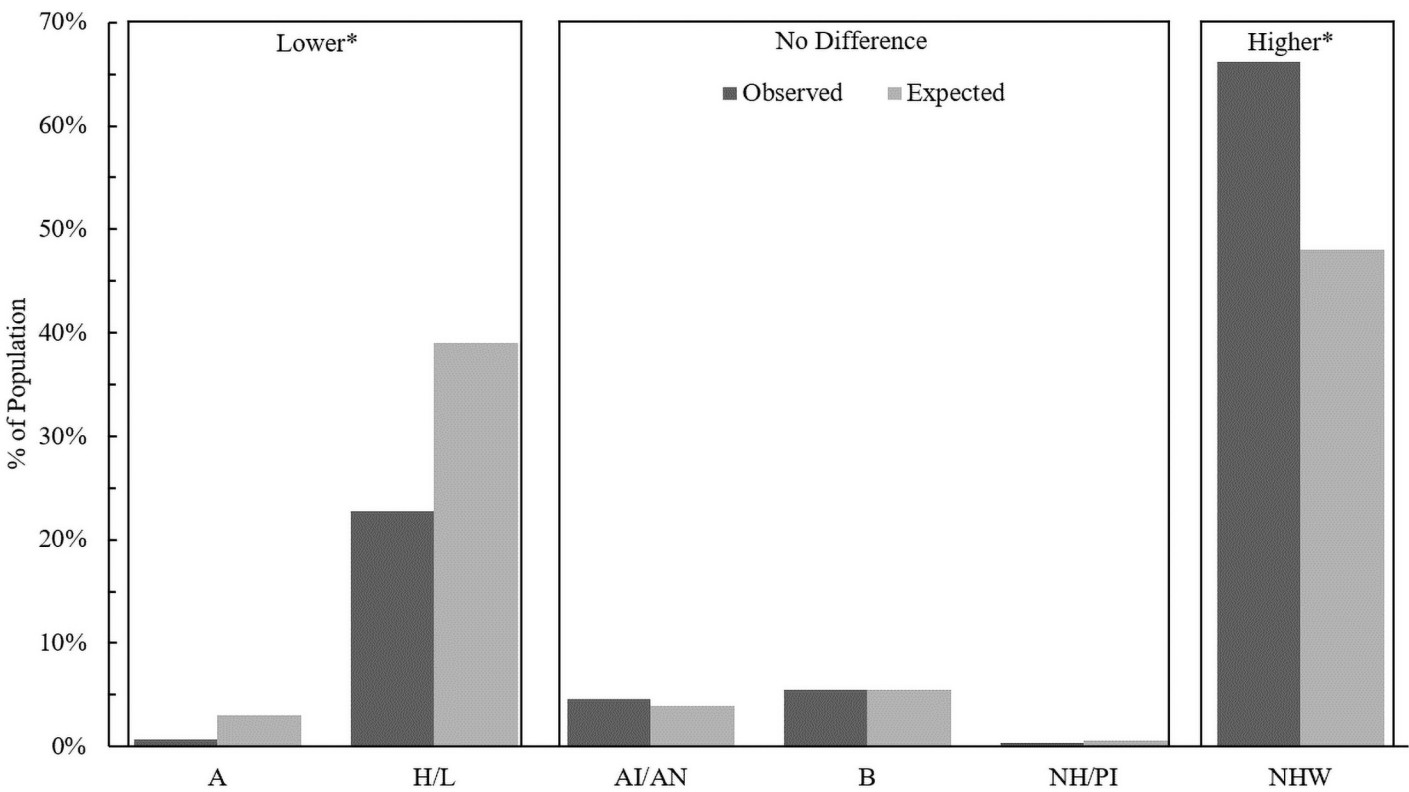

**Fig 3. Comparison of the observed versus expected proportions of opioid dependence at the time of giving birth in each racial/ethnic group of mothers.** Boxes labelled higher or lower mean that the observed proportions are significantly higher or lower than the expected proportion after post-hoc comparisons that incorporate a Bonferroni correction with six groups (p<0.001/6). A: Asian, H/L: Hispanic or Latino, AI/AN: American Indian or Alaskan Native, B: Black, NH/PI: Native Hawaiian or Pacific Islander, NHW: Non-Hispanic White.

composed of tribal nations. Rather, we have reported those PCAs back to the ADHS for use in their decision processes. An initial chi-square test revealed that opioid dependence among mothers who had given birth significantly deviated from the expected distribution across PCAs. Post-hoc comparisons revealed that there were significantly more mothers who were dependent on opioids residing in the following PCAs than expected: Casas Adobes, Encanto Village, Flowing Wells, Globe, North Mountain Village, Prescott, Safford, Kingman, Tucson Central, Tucson East, Tucson Foothills, and Tucson South (Fig 7). The following PCAs had significantly fewer mothers who were dependent on opioids at the time of giving birth than expected: Buckeye, Estrella Village and Tolleson, Gilbert Central, Gilbert South, Maryvale Village, and Yuma. Future studies may investigate which PCA characteristics may contribute to or mitigate opioid dependence among pregnant women and women of child-bearing age.

## Associated comorbid conditions

As mentioned in the Methods section, we used random forests to select comorbidities associated with NOWS and opioid dependence. Presence or absence of an ICD9 or 10 code for NOWS or opioid dependence was used as the labelled target variable. We analyzed four sets of data for identification of comorbid conditions: Neonates with ICD9 codes, Neonates with ICD10 codes, Mothers with ICD9 codes, Mothers with ICD10 codes. For neonates with NOWS (Table 2), we found that, in agreement with previous studies, feeding problems, respiratory distress (transitory tachypnea), and neonatal jaundice commonly co-occurred with

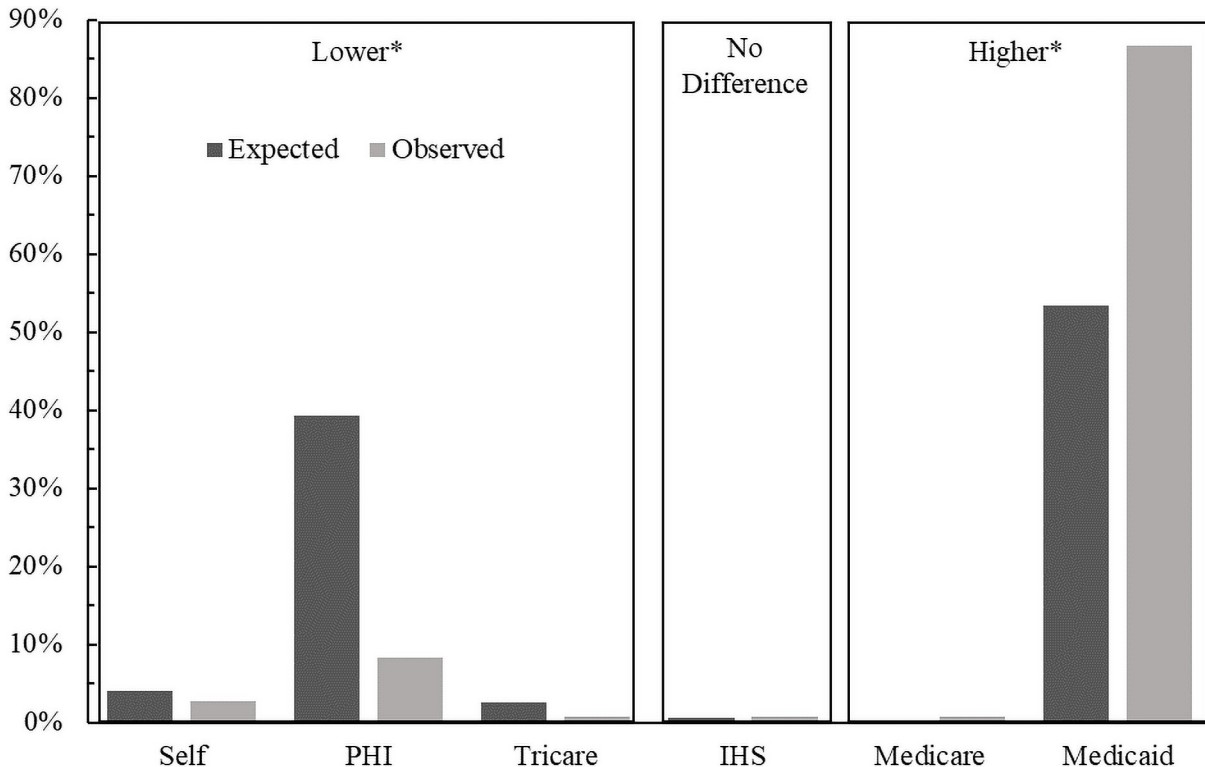

**Fig 4. Comparison of the observed versus expected proportions of NOWS at the time birth in each payor group utilized for infants.** Boxes labelled higher or lower mean that the observed proportions are significantly higher or lower than the expected proportion after post-hoc comparisons that incorporate a Bonferroni correction with six groups (p<0.001/6). PHI: Private Health Insurance, Self: Self Pay, IHS: Indian Health Services.

NOWS [27–29]. We also found that neonatal candidiasis infection and diaper or skin rash (diaper dermatitis) were among top-ranked comorbid conditions.

Analysis of mothers' discharge records (Table 3) identified several comorbid conditions in agreement with previous research on women with opioid dependence during pregnancy. Poly-substance use, notably tobacco, alcohol, and stimulants, is more common among pregnant women who use opioids [30]. Chronic pain, mental health conditions, unspecified anxiety, and other viral illnesses were also highly-ranked comorbid conditions.

## Discussion

Neonates with NOWS in Arizona and their mothers from 2010–2017 tended to be socioeconomically disadvantaged, non-Hispanic White, and geographically clustered throughout Arizona. In addition, mothers in this group are unpartnered more frequently than expected, which may indicate a relative lack of social support. Unsurprisingly, characteristics of mothers with opioid dependence at the time of birth were closely related to those of neonates with NOWS. There does not appear to be an increase in reported of maternal opioid dependence from 2016 to 2017. Mothers who were opioid dependent accumulated nearly $8000 more in total charges and stayed a half day longer than those who were not opioid dependent. There does not appear to be a significant age difference between mothers with opioid dependence at the time of birth and those who are not opioid dependent.

There are major inconsistencies in substance use screening and NOWS diagnosis and treatment [31, 32]. The American Academy of Pediatrics has called for more similarity and

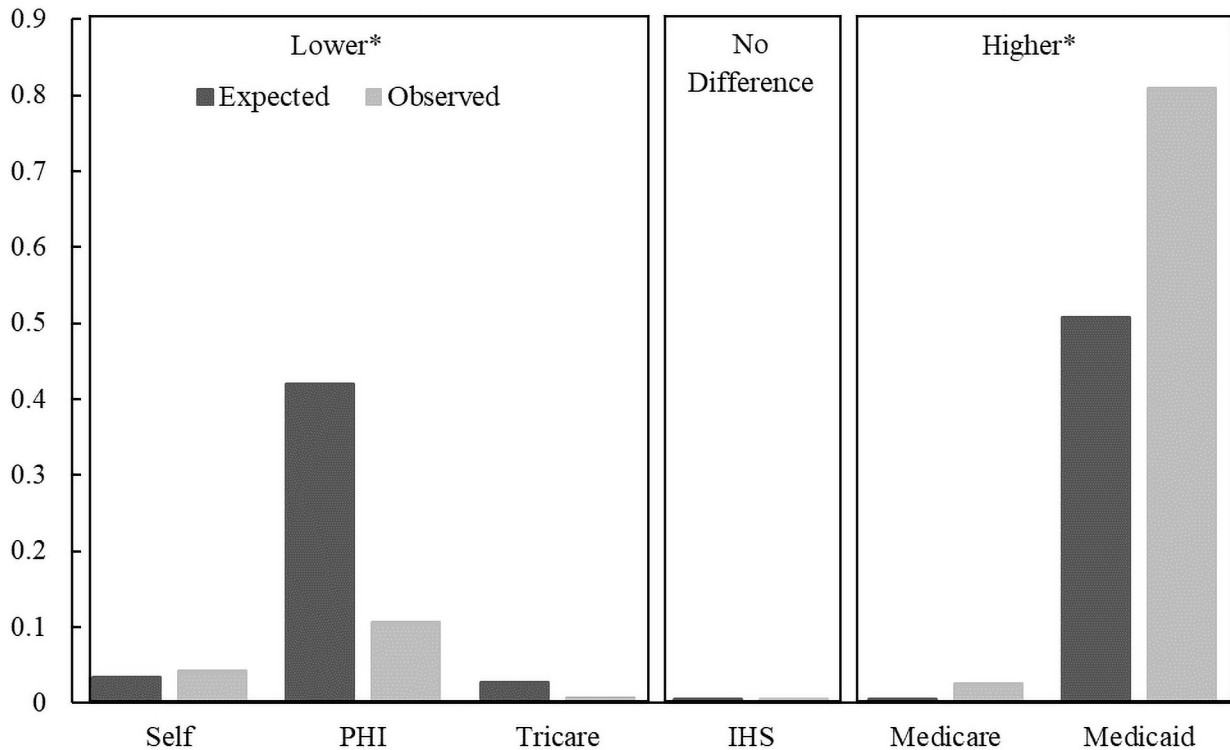

**Fig 5. Comparison of the observed versus expected proportions of opioid dependence at the time of giving birth in each payor group utilized for mothers.** Boxes labelled higher or lower mean that the observed proportions are significantly higher or lower than the expected proportion after post-hoc comparisons that incorporate a Bonferroni correction with six groups (p<0.001/6). PHI: Private Health Insurance, Self: Self Pay, IHS: Indian Health Services.

standardization of care for neonates with NOWS [33]. Standardized screening and treatment also has the potential to improve care for neonates with NOWS and their mothers who are opioid dependent at the time of giving birth [31, 34]. In an effort to improve and standardize NOWS treatment, there is a need for better and more reliable criteria for screening and early identification of NOWS cases. Analysis of hospital discharge records is one step toward better characterization of co-morbid conditions that could improve precision in early NOWS identification.

In a recent study of vaginal flora, Farr et al observed significantly higher rates of candidiasis in pregnant mothers receiving medication assisted opioid treatment than in control groups [35]. Diaper dermatitis a known condition common among neonates with NOWS, however, it is not considered a reliable diagnostic criterion [36]. Our data suggest that considerably more attention should be paid to potential links between dermatitis and vaginal candidiasis with NOWS. To our knowledge there are no studies linking increased rates of neonatal candidiasis with vaginal candidiasis increases among mothers using medication assisted (opioid maintenance) treatment.

Like the neonate analysis, analysis of mothers' discharge records identified several comorbid conditions that previous studies have found to be associated with opioid dependence during pregnancy. Polysubstance use is more common among pregnant women who use opioids [30]. Tobacco use is particularly prevalent among this population, with estimates of tobacco use as high as 85–90% among pregnant women treated with buprenorphine or methadone [37–39]. Pregnant women using opioids have been found to be more likely to be diagnosed with depression, anxiety, post-traumatic stress disorder, and panic disorder [40], which is also

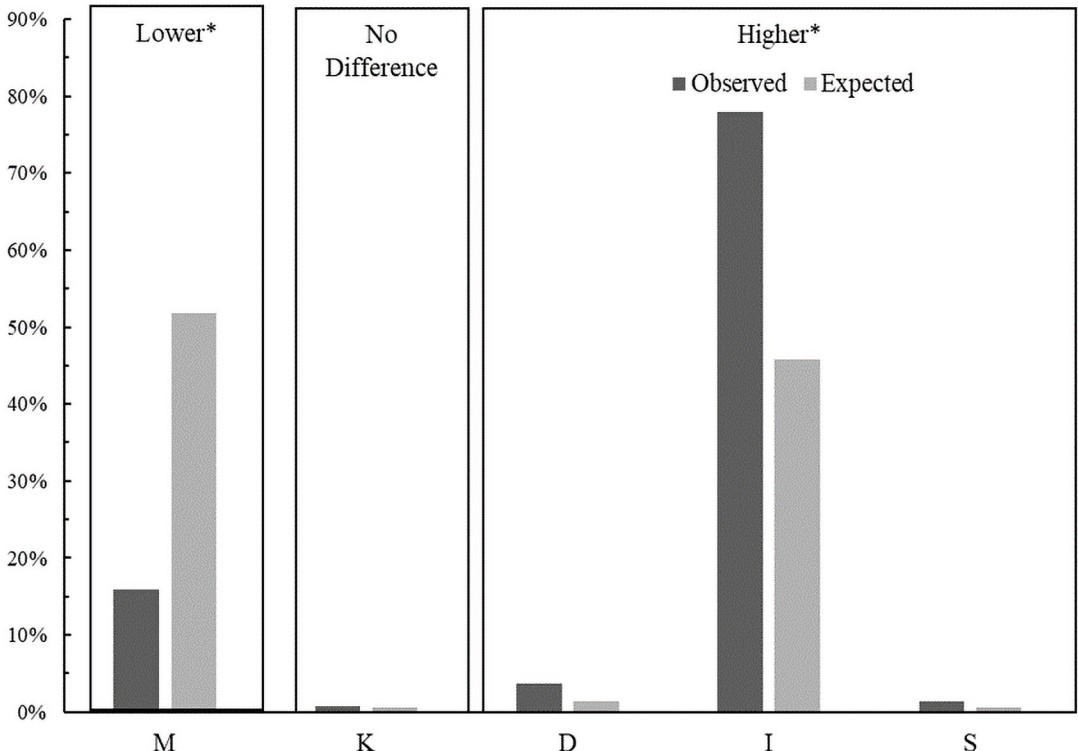

**Fig 6. Comparison of the observed versus expected proportions of opioid dependence at the time of giving birth in by maternal marital status.** Boxes labelled higher or lower mean that the observed proportions are significantly higher or lower than the expected proportion after post-hoc comparisons that incorporate a Bonferroni correction with six groups (p<0.001/5). M: Married, K: Unknown, D: Divorced, I: Single, S: Separated.

in agreement with our study, where maternal mental disorders are highly ranked. Interestingly, while we found studies that identified increased opioid prescriptions written to women with perineal lacerations [41], to our knowledge no studies have reported an association between perineal lacerations and opioid dependence at the time of giving birth. Epidural anesthesia has been associated with increased perineal laceration [42] and mothers who are opioid dependent at the time of giving birth are not candidates for pain management with opioids [43, 44]. Future research should consider whether the risk of perineal laceration among opioid dependent women is elevated due to pain management practices or other factors.

Although additional studies are necessary to better understand the association between marital status and opioid use disorder, we suspect that marital status is a proxy for social support. Previous studies reported that married individuals are less likely to use illicit drugs [45] and those who participate in substance-abuse treatment programs are more likely to experience positive outcomes [46–49]. Heinz et al. found that that close spousal relationships were a good predictor of reduced cocaine and heroin use in individuals during and after treatment [50]. With the results of these previous studies, our results suggest further investigation into outcomes associated with marital and perhaps other forms of social support when considering opioid use disorder in pregnant women and women of childbearing age.

Areas with higher than expected rates of NOWS in relation to population estimates warrant additional research. In our meetings with stakeholders and investigation of local understandings of areas with higher rates of drug use in the state, observations were consistent with our findings. For example, the Prescott area is known for a proliferation of "sober living houses" in

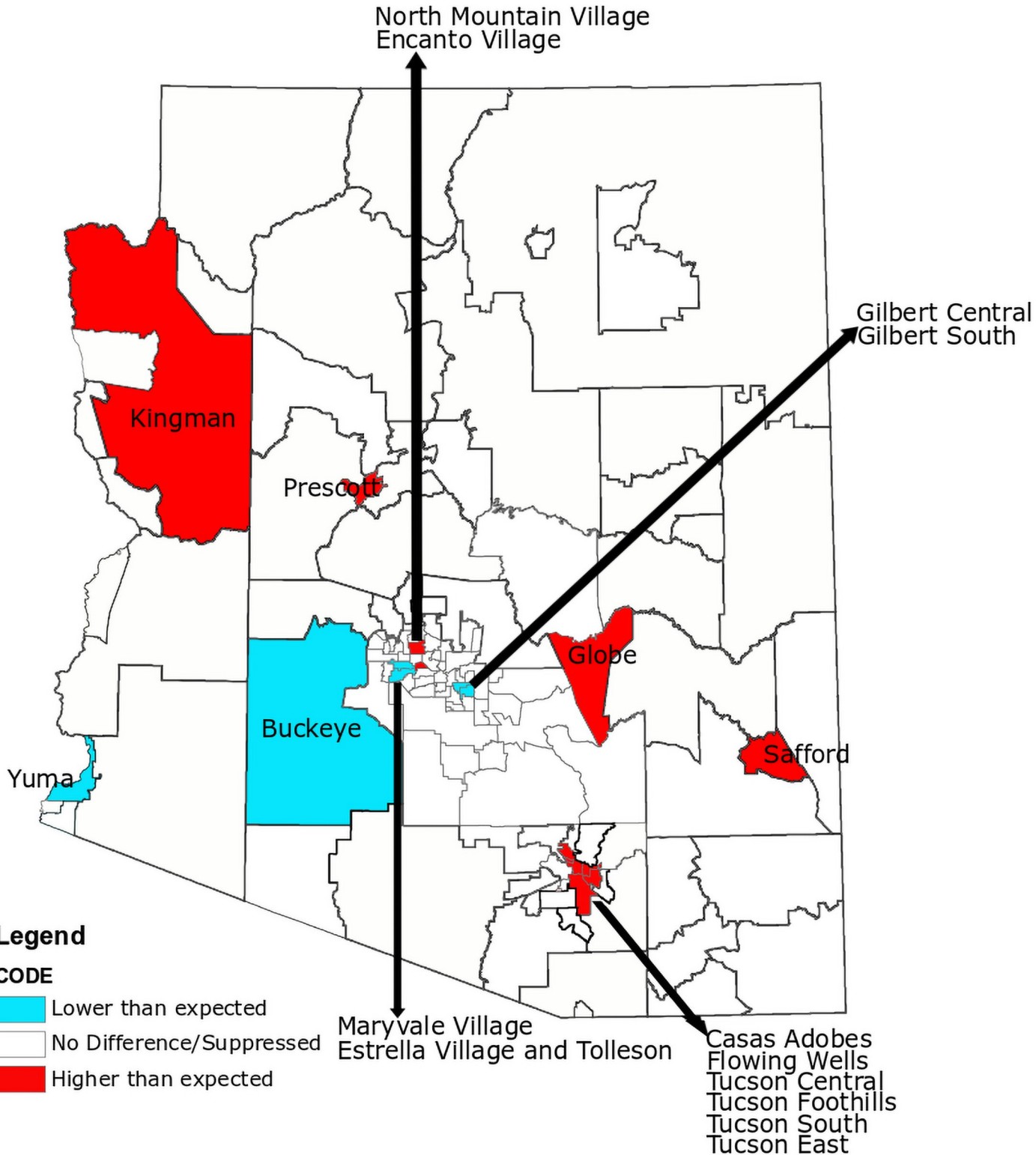

**Fig 7. Comparison of the observed versus expected proportions of opioid dependence at the time of giving birth by maternal residential PCA.** Red or blue indicates that there is a significantly higher or lower number of mothers using opioids than expected. A map with all PCAs labelled can be found on the ADHS website: https://www.azdhs.gov/documents/prevention/health-systems-development/data-reports-maps/maps/azpca.pdf.

**Table 2. Top ranked ICD9 and ICD10 codes associated with infants who have NOWS.**

| Rank | ICD9 Code | ICD10 Code (and Rank) | Description |
|------|-----------|------------------------|-------------|
| 1 | 779.5 | P961 (1) | Drug withdrawal syndrome in newborn |
| 2 | 779.31 | P92.1–2 and 8–9 (4) | Feeding problems in newborn |
| 3 | 760.72 | P04.49 (2) | Hallucinogenic agents affecting fetus or newborn via placenta or breast milk |
| 4 | 7746 | P599 (5) | Unspecified fetal and neonatal jaundice |
| 5 | 6910 | L22 (3) | Diaper or napkin rash |
| 6 | 770.6 | P22.1 (6) | Transitory tachypnea of newborn |
| 7 | 760.79 | P04.9 (9*) | Other noxious influences affecting fetus or newborn via placenta or breast milk |
| 8 | 771.7 | P37.5 (7) | Neonatal Candida infection |
| 9 | V290 | P002 (13) | Observation for suspected infectious condition |
| 10 | V05.3 | Z23 (19) | Need for prophylactic vaccination and inoculation against viral hepatitis |
| 11 | V30.00 | Z38.00 (8) | Single liveborn, born in hospital, delivered without mention of cesarean section (i.e. delivered vaginally) |
| 16 | 745.4 and 745.6 | Q21.1 (10) | Atrial septal defect |

ICD10 codes were used for ranking infants admitted after Oct. 1 2015, as well as for any infants born in health care facilities that adopted ICD10 codes prior to Oct. 1, 2015. Conditions are in the ICD9 rank order (see first column), and the corresponding ICD10 and rank of the ICD10 code are listed. The increase from rank 11 to rank 16 was allowed so that all top 10 ICD10 codes could be shown.

recent years, with so many recovering substance users coming to the area from outside that the trend has been reported in state and national news outlets [51, 52]. Low numbers of medication assisted treatment providers in areas with the highest rates of opioid use and NOWS cases are also potentially responsible for higher than expected rates in some, particularly rural, areas of the state [53].

## Limitations

This analysis of hospital discharge records is limited by reliance on secondary data for which reporting may be inconsistent. We expect that heightened surveillance implemented during the study period may have improved consistency of reporting, however, we can only comment on what is recorded at discharge. Further, length of stay and total charges can vary based on treatment approaches, early identification, and hospital policies. It is outside the scope of this analysis to determine what approaches were used in Arizona NOWS cases or what impact those practices had on NOWS cases in Arizona.

Tribal PCAs were not included in this analysis due to data restrictions and non-reporting. The exclusion of Tribal PCAs limits characterization of racial/ethnic demographics and may not include accurate accounting of a Native American/Alaska Native populations who experience the highest incidence rates of NOWS nationwide [54].

## Conclusions

The comorbidity analysis, using supervised machine learning, revealed that diaper dermatitis and jaundice were more importantly associated with NOWS than traditional conditions of respiratory distress and irritability. Additionally, to our knowledge, neonatal candidiasis has

**Table 3. Top ranked ICD9 and ICD10 codes associated with mothers who are opioid dependent at the time of giving birth.**

| Rank | ICD9 Code | ICD10 Code (and Rank) | Description |
|---|---|---|---|
| 1 | 304 | F11.20, F11.90, F1110 (1,3,4) | Opioid type dependence, unspecified |
| 2 | 648.31 | O99.324 (2) | Drug dependence of mother, delivered, with or without mention of antepartum condition |
| 3 | 648.41 | O99.344 (8) | Mental disorders of mother, delivered, with or without mention of antepartum condition |
| 4 | 649.01 | O99.334 and F17.210 (5,6) | Tobacco use disorder complicating pregnancy, childbirth, or the puerperium, delivered, with or without mention of antepartum condition |
| 5 | 338.29 | G89.29 (9) | Other chronic pain |
| 6 | V23.7 | O09.30 (7) | Supervision of high-risk pregnancy with insufficient prenatal care |
| 7 | 644.21 | O60.14X0 (57) | Early onset of delivery, delivered, with or without mention of antepartum condition |
| 8 | 648.91 | No specific conversion | Other current conditions classifiable elsewhere of mother, delivered, with or without mention of antepartum condition |
| 9 | 647.61 | O98.42, 098.511–513, O98.52 (31) | Other viral diseases in the mother, delivered, with or without mention of antepartum condition |
| 10 | 300 | F41.9 (15) | Anxiety state, unspecified |
| 11 | 664.11 | O70.1 (14) | Second-degree perineal laceration, delivered, with or without mention of antepartum condition |
| 12 | 656.51 | O36.5110–130, O36.5910–30 (26) | Poor fetal growth, affecting management of mother, delivered, with or without mention of antepartum condition |
| 62 | 305.70–72 | F15.10 (10) | Amphetamine or related acting sympathomimetic abuse (unspecified, continuous, or episodic) |

ICD10 codes were used for ranking mothers admitted after Oct. 1 2015, as well as for any infants born in health care facilities that adopted ICD10 codes prior to Oct. 1, 2015. Conditions are in the ICD9 rank order (see first column), and the corresponding ICD10 and rank of the ICD10 code are listed. The increase from rank 12 to rank 62 was allowed so that the top 10 ICD10 codes could be shown.

not been previously associated with NOWS. However, our results, coupled with the finding that pregnant women receiving medication assisted opioid treatment are more frequently colonized with *Candida* [35], suggests that it may be important to screen pregnant women receiving medication assisted opioid treatment for candidiasis to develop maternal and neonatal treatment strategies. Our analysis of the Arizona Hospital Discharge Database from 2010 to 2017 suggests a need for better characterization of comorbid conditions in NOWS neonates and their mothers. Variability of comorbid conditions in NOWS neonates suggest that much more detailed understanding of contributors to and symptoms of NOWS could begin to address challenges in standardization of care. Efforts to improve treatment as well as primary and secondary prevention of NOWS may benefit from better characterization of comorbid and frequently co-occurring conditions. Future research should also consider potential interactions between opioid exposure and comorbid conditions in utero.

## Supporting information

**S1 Table. Summary of non-geographic data used in this study by figure.** The Mothers[O] column contains data for all mothers who were dependent on opioids at the time of giving birth. The Infants[NOWS] column contains data for all infants who had NOWS. Totals at the end of each category are not the same for all categories due to missing data or because some patients reported atypical categories (e.g. payer was workers compensation or individual was a foreign national). (DOCX)

## Acknowledgments

The authors acknowledge the contributions of Arizona Department of Health Services staff members Kyle Gardner and Timothy Flood for assistance defining the data request, the

Northern Arizona University Information Technology Services for managing data security issues, and the Southwestern Health Equity Research Collaborative Research Infrastructure Core for analytical assistance.

## Author Contributions

**Conceptualization:** Emery R. Eaves, Jarrett Barber, Sara A. Clancey, Joseph Spadafino, Crystal M. Hepp.

**Data curation:** Ryann Whealy, Joseph Spadafino, Crystal M. Hepp.

**Formal analysis:** Jarrett Barber, Ryann Whealy, Sara A. Clancey, Jill Hager Cocking, Joseph Spadafino, Crystal M. Hepp.

**Funding acquisition:** Emery R. Eaves, Sara A. Clancey, Crystal M. Hepp.

**Investigation:** Emery R. Eaves, Jarrett Barber, Sara A. Clancey, Rita Wright, Jill Hager Cocking, Crystal M. Hepp.

**Methodology:** Jarrett Barber, Rita Wright, Crystal M. Hepp.

**Project administration:** Crystal M. Hepp.

**Supervision:** Crystal M. Hepp.

**Visualization:** Crystal M. Hepp.

**Writing – original draft:** Emery R. Eaves, Jarrett Barber, Sara A. Clancey, Rita Wright, Crystal M. Hepp.

**Writing – review & editing:** Emery R. Eaves, Jarrett Barber, Ryann Whealy, Sara A. Clancey, Rita Wright, Jill Hager Cocking, Joseph Spadafino.

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
