## [Decision Letter · Decision Letter 0]

14 Oct 2020

PONE-D-20-17306

Characterization of Neonatal Abstinence Syndrome in Arizona from 2010-2017

PLOS ONE

Dear Dr. Hepp,

Thank you for submitting your manuscript to PLOS ONE. After careful consideration, we feel that it has merit but does not fully meet PLOS ONE’s publication criteria as it currently stands. Therefore, we invite you to submit a revised version of the manuscript that addresses the points raised during the review process.

We look forward to receiving your revised manuscript.

Kind regards,

Barbara Wilson Engelhardt, MD

Academic Editor

PLOS ONE

Journal Requirements:

2. In the ethics statement in the manuscript and in the online submission form, please provide additional information about the patient records used in your retrospective study.

Specifically, please ensure that you have discussed whether all data were fully anonymized before you accessed them and/or whether the IRB or ethics committee waived the requirement for informed consent.

If patients provided informed written consent to have data from their medical records used in research, please include this information.

3. In your Methods section, please provide additional information about the methodology used, for example by listing the comorbidities analysed, and describing how variables were defined and categorised.

5. Please ensure that you refer to Figures 6 and 7 in your text as, if accepted, production will need this reference to link the reader to the figure.

6. We note you have included a table to which you do not refer in the text of your manuscript. Please ensure that you refer to Table 3 in your text; if accepted, production will need this reference to link the reader to the Table.

Reviewers' comments:

Reviewer's Responses to Questions

**Comments to the Author**

1. Is the manuscript technically sound, and do the data support the conclusions?

Reviewer #1: Partly

Reviewer #2: Yes

2. Has the statistical analysis been performed appropriately and rigorously? 

Reviewer #1: Yes

Reviewer #2: Yes

3. Have the authors made all data underlying the findings in their manuscript fully available?

Reviewer #1: Yes

Reviewer #2: Yes

4. Is the manuscript presented in an intelligible fashion and written in standard English?

Reviewer #1: Yes

Reviewer #2: Yes

5. Review Comments to the Author

Reviewer #1: RE: Review of PONE-D-20-17306; Characterization of Neonatal Abstinence Syndrome in Arizona from 2010-2017

DATE: August 21, 2020

Thank you for the opportunity to review this manuscript, which summarizes the occurrence of NAS over 8 years in Arizona. The data are unique in that the they span before and after the installation of a NAS surveillance system. The data also end in 2017, the year that the AZ governor declared that opioid use disorder had reached epidemic levels. I believe there is great potential for this paper, but have recommendations that may enhance its quality. I will review these in the order that they appear.

ABSTRACT

The structure of the abstract should be revisited so that the lit review is completed prior to stating the purpose of the study. Following the research justification and purpose, the data description and results would follow. Currently, after the research purpose, the authors revisit literature justification prior to moving into the results. This is an awkward sequencing.

INTRODUCTION

Line 56: Treatment protocol for NAS does not seem relevant for this paper unless you wish to link the procedure to length of stay. Clarify the importance of treatment in relation to the study or delete.

Line 69-76: It is unclear how the discussion of prevention, screening for NAS, and standardization of care for infants with NAS is relevant to this manuscript. These issues may be worked into the discussion perhaps but they do not assist in justifying the importance of the study. I recommend the deletion of this paragraph. More attention could be given to what we know about the demographics of mothers with infants with NAS, and their comorbidities, nationally or in other states.

Line 75: Note, the early identification of NAS cases can be challenging as some infants do not exhibit signs of withdrawal until days after their birth.

Line 93: Clarify why the initiation of the surveillance system would lead to a hypothesized increase in NAS cases. Given that NAS is determined based on the medical codes, the codes should remain consistent for billing purposes – both before and after the surveillance system. Based on existing data, it seems this increase was typical, until greater efforts to intervene were implemented.

METHODS

Line 142: The study includes only non-tribal PCAs. In attempts to better understand the state of AZ, the exclusion of the tribal areas diminishes the ability to address the study purpose. The possible impact of not including tribal areas should be explored further in the Discussion, based on what is understood about NAS in the tribal communities.

RESULTS and DISCUSSION

Line 164: The cases of NAS are not clearly laid out to test whether there was an increase in cases after the surveillance system was implemented. As a proposed hypothesis this should be explicitly tested. (e.g., in relation to Figure 1).

Line 191: The race, ethnicity, marriage status, and insurance coverage are very comparable to the data across the US. Ideally such data from earlier studies would be reflected in the Introduction.

Length of stay can vary greatly in whether infants are released earlier to complete weaning at home, use of rescue dosing, and the inclusion or exclusion of complementary treatment approaches. The variability of the AZ treatment approaches should be discussed in the Discussion.

Line 262: This is an example of the Discussion lurking into the Results. This is not in keeping with manuscript sections. Please keep the Results in the first section and the Discussion points would then follow.

Line 296: Given the variability of the comorbid conditions, it seems that this may contribute to challenges related to standardization of care. This needs to be revisited and better clarified.

In sum, I believe these data have much to contribute to our understanding of NAS in relation to both infants and their mothers. However, opportunities to strengthen this contribution remain. Toward this end, I hope these comments are beneficial

Reviewer #2: Comments to author

Excellent study that contributes to the evidence. Minor revisions are outlined for your consideration. Recommend consideration of replacing infants with neonates. There were inconsistencies in formatting regarding indention of new paragraphs, I pointed these out initially.

Page 8 Abstract

Line 38 & 41 suggest separating ethnic/racial with a comma as ethnic, racial, and/or..

Line 44 consider rephrasing to best outline the withdrawal of substances such as “infants opioid exposed in utero who experience withdrawal following birth.”

Line 45 rephrase the portion of this sentence “less well understood” consider “there remains a gap in regional trends.”

Line 46 rephrase portion of the sentence “we find that..” consider “Our findings suggest…”

Line 48 delete “we find that”

Line 50 consider rephrasing “we report…” to “we identified comorbidities associated with … not previously reported

Introduction

Line 57 in discussing treatment of NAS, consider mentioning non-pharmacologic interventions as the first line therapy

Line 61 Consider rewording “identify” and discuss diagnostic criteria

Line 65 rephrase the sentence “we found..” I had to read this a few times to understand the message and it is a vital message

Line 67 “improving understanding in these areas..” state what you are referring to for the reader – comorbid conditions?

Line 70 “changes in the prescribing of opioid drugs” could be rephrased such as “opioid medication modifications and routine screening for substance use.”

Line 71 ..and the American.. consider adding period in that prior sentence and start a new sentence stating AAP recommendations

Line 74 consider rephrasing “Better and …” such as “In an effort to improve and standardized NAS treatment, there is a need for xxx.”

Page 10

Line 123 consider replacing newborn infants with neonates

Line 157 delete “very small”

Page 11 Results and Discussion consider separating these two sections out

Line 164 indent, consider rephrasing this sentence such as “The incidence of NAS in Arizona continues to rise..”

Line 166 consider rephrasing your sentence on “the large number of infants… “ recommend separating this point from the sentence and highlight the incidence of NAS is a sequela/consequence from the opioid epidemic

172 Indent, again consider neonate rather than “newborn infant” and rephrase this sentence for clarity.

181 Indent. Curious are these rates including infants treated in the intensive care, meaning are these NAS neonates of all gestational ages compared to all conditions/neonates. Would be a great point to highlight if so..

191 Indent.

205 Rephrase “past studies,” consider “Previous studies reported…”

Page 12

207 delete duplicate that

232 consider rephrasing “we are planning..” consider “future studies may investigate..”

Page 13

264 Discussion/Conclusions – Consider deleting discussion

6. PLOS authors have the option to publish the peer review history of their article (what does this mean?). If published, this will include your full peer review and any attached files.

Reviewer #1: No

Reviewer #2: No

---

## [Author Response · Author response to Decision Letter 0]

6 Jan 2021

Journal Requirements:

AUTHORS’ RESPONSE: We have carefully reformatted according to PLOS ONE’s formatting requirements as shown in the linked documents.

2. In the ethics statement in the manuscript and in the online submission form, please provide additional information about the patient records used in your retrospective study.

Specifically, please ensure that you have discussed whether all data were fully anonymized before you accessed them and/or whether the IRB or ethics committee waived the requirement for informed consent.

If patients provided informed written consent to have data from their medical records used in research, please include this information.

AUTHORS’ RESPONSE: This effort was deemed to be public health surveillance and epidemiological activity, and not research, by both the Arizona Department of Health Services Human Subjects Research Board and Northern Arizona University Institutional Review Board. Because it is not research, the limited dataset, which was deidentified to the federal standards of a limited dataset associated with a data use agreement (between the Arizona Department of Health Services and Northern Arizona University) prior to us receiving it, did not require participant consent.

3. In your Methods section, please provide additional information about the methodology used, for example by listing the comorbidities analysed, and describing how variables were defined and categorised.

AUTHORS’ RESPONSE: We added language in the METHODS – Associated Comorbid Conditions section to highlight that all possible diagnosis codes (~13,000 IDC-9-CM and ~70,000 ICD-10-CM possible codes) from each patient record are assessed for their potential to be a comorbid condition.

AUTHORS’ RESPONSE: The limited data set that was accessed by our team was made available to us because the effort was classified as public health surveillance and not research. We are unsure of how the Arizona Department of Health Services would choose to classify an identical data request, as the classification of research versus public health surveillance depends on a researcher’s motivation as well as whether the researcher is collaborating with the public health agency for a public health purpose. Additionally, the data set that we received was considered to be fully deidentified due to the data use agreement that Northern Arizona University has with the Arizona Department of Health Services. However, it was provided to us as a limited use, rather than public use, data set. As part of our data request, we are only to report data in aggregate format (as in Table S1). Researchers wishing to reproduce or build on this study will need to submit a data request to the Arizona Department of Health Services to be approved: https://www.azdhs.gov/documents/director/administrative-counsel-rules/HSRB_NewProductSubmission.pdf

We have also added our response to the revised cover letter as requested.

AUTHORS’ RESPONSE: Response is provided under 4a (directly above).

5. Please ensure that you refer to Figures 6 and 7 in your text as, if accepted, production will need this reference to link the reader to the figure.

AUTHORS’ RESPONSE: We have added references to Figures 6 and 7 to the text. 

6. We note you have included a table to which you do not refer in the text of your manuscript. Please ensure that you refer to Table 3 in your text; if accepted, production will need this reference to link the reader to the Table.

AUTHORS’ RESPONSE: We have corrected this omission and Table 3 is now listed appropriately in the text.

Reviewers' comments:

Reviewer's Responses to Questions

Comments to the Author

1. Is the manuscript technically sound, and do the data support the conclusions?

Reviewer #1: Partly

Reviewer #2: Yes

2. Has the statistical analysis been performed appropriately and rigorously?

Reviewer #1: Yes

Reviewer #2: Yes

3. Have the authors made all data underlying the findings in their manuscript fully available?

Reviewer #1: Yes

Reviewer #2: Yes

4. Is the manuscript presented in an intelligible fashion and written in standard English?

Reviewer #1: Yes

Reviewer #2: Yes

5. Review Comments to the Author

Reviewer #1: RE: Review of PONE-D-20-17306; Characterization of Neonatal Abstinence Syndrome in Arizona from 2010-2017

DATE: August 21, 2020

Thank you for the opportunity to review this manuscript, which summarizes the occurrence of NAS over 8 years in Arizona. The data are unique in that the they span before and after the installation of a NAS surveillance system. The data also end in 2017, the year that the AZ governor declared that opioid use disorder had reached epidemic levels. I believe there is great potential for this paper, but have recommendations that may enhance its quality. I will review these in the order that they appear.

AUTHORS’ RESPONSE: We appreciate the reviewers’ careful review and helpful suggestions. We have detailed edits and how we addressed each comment below.

ABSTRACT

The structure of the abstract should be revisited so that the lit review is completed prior to stating the purpose of the study. Following the research justification and purpose, the data description and results would follow. Currently, after the research purpose, the authors revisit literature justification prior to moving into the results. This is an awkward sequencing.

AUTHORS’ RESPONSE: We have restructured the abstract based on the reviewers’ suggestions. 

INTRODUCTION

Line 56: Treatment protocol for NAS does not seem relevant for this paper unless you wish to link the procedure to length of stay. Clarify the importance of treatment in relation to the study or delete.

Line 69-76: It is unclear how the discussion of prevention, screening for NAS, and standardization of care for infants with NAS is relevant to this manuscript. These issues may be worked into the discussion perhaps but they do not assist in justifying the importance of the study. I recommend the deletion of this paragraph. More attention could be given to what we know about the demographics of mothers with infants with NAS, and their comorbidities, nationally or in other states.

AUTHORS’ RESPONSE: We have deleted the sentence about treatment protocols and moved the lines about standardization in screening to the discussion (page 10). We hope this change addresses both of the reviewers’ above comments. 

Line 75: Note, the early identification of NAS cases can be challenging as some infants do not exhibit signs of withdrawal until days after their birth.

AUTHORS’ RESPONSE: We have deleted this paragraph from the introduction and clarified how early identification relates to the purpose of this paper in the discussion (page 10). 

Line 93: Clarify why the initiation of the surveillance system would lead to a hypothesized increase in NAS cases. Given that NAS is determined based on the medical codes, the codes should remain consistent for billing purposes – both before and after the surveillance system. Based on existing data, it seems this increase was typical, until greater efforts to intervene were implemented.

AUTHORS’ RESPONSE: We have changed this in response to the reviewers’ comment and the text now reflects the more descriptive nature of this analysis in the context of increased surveillance (Page 2).

METHODS

Line 142: The study includes only non-tribal PCAs. In attempts to better understand the state of AZ, the exclusion of the tribal areas diminishes the ability to address the study purpose. The possible impact of not including tribal areas should be explored further in the Discussion, based on what is understood about NAS in the tribal communities.

AUTHORS’ RESPONSE: We have added a discussion of this in the limitation sections that we added after the newly created discussion question and linked it to what is known nationwide.

RESULTS and DISCUSSION

Line 164: The cases of NAS are not clearly laid out to test whether there was an increase in cases after the surveillance system was implemented. As a proposed hypothesis this should be explicitly tested. (e.g., in relation to Figure 1).

AUTHORS’ RESPONSE: Based on the reviewers’ comment, we removed the hypothesis that there would be an increase in cases. 

Line 191: The race, ethnicity, marriage status, and insurance coverage are very comparable to the data across the US. Ideally such data from earlier studies would be reflected in the Introduction.

Length of stay can vary greatly in whether infants are released earlier to complete weaning at home, use of rescue dosing, and the inclusion or exclusion of complementary treatment approaches. The variability of the AZ treatment approaches should be discussed in the Discussion.

AUTHORS’ RESPONSE: We have added more literature review to reflect data from earlier studies in the introduction (page 2). The variability of Arizona treatment approaches is outside the scope of this analysis because we are focused on hospital discharge records, which report diagnoses. We have added a section on limitations to note that this is a limitation. 

Line 262: This is an example of the Discussion lurking into the Results. This is not in keeping with manuscript sections. Please keep the Results in the first section and the Discussion points would then follow.

AUTHORS’ RESPONSE: We have separated out these instances of discussion into a separate discussion section in response to the reviewers’ comment. 

Line 296: Given the variability of the comorbid conditions, it seems that this may contribute to challenges related to standardization of care. This needs to be revisited and better clarified.

AUTHORS’ RESPONSE: We have revisited this and attempted to clarify in the conclusion (page 11). 

In sum, I believe these data have much to contribute to our understanding of NAS in relation to both infants and their mothers. However, opportunities to strengthen this contribution remain. Toward this end, I hope these comments are beneficial

AUTHORS’ RESPONSE: Thank you for this thorough review. We have attempted to address each comment carefully. 

Reviewer #2: Comments to author

Excellent study that contributes to the evidence. Minor revisions are outlined for your consideration. Recommend consideration of replacing infants with neonates. There were inconsistencies in formatting regarding indention of new paragraphs, I pointed these out initially.

AUTHORS’ RESPONSE: We have changed “infant” to “neonate” throughout and corrected formatting issues per PLoS One’s guidelines. 

Page 8 Abstract

Line 38 & 41 suggest separating ethnic/racial with a comma as ethnic, racial, and/or..

Line 44 consider rephrasing to best outline the withdrawal of substances such as “infants opioid exposed in utero who experience withdrawal following birth.”

Line 45 rephrase the portion of this sentence “less well understood” consider “there remains a gap in regional trends.”

Line 46 rephrase portion of the sentence “we find that..” consider “Our findings suggest…”

Line 48 delete “we find that”

Line 50 consider rephrasing “we report…” to “we identified comorbidities associated with … not previously reported

AUTHORS’ RESPONSE: We have made all of the suggested edits about to the abstract. We appreciate the reviewers’ attention to detail and suggestions that we hope have increased clarity. 

Introduction

Line 57 in discussing treatment of NAS, consider mentioning non-pharmacologic interventions as the first line therapy

Line 61 Consider rewording “identify” and discuss diagnostic criteria

Line 65 rephrase the sentence “we found..” I had to read this a few times to understand the message and it is a vital message

Line 67 “improving understanding in these areas..” state what you are referring to for the reader – comorbid conditions?

Line 70 “changes in the prescribing of opioid drugs” could be rephrased such as “opioid medication modifications and routine screening for substance use.”

Line 71 ..and the American.. consider adding period in that prior sentence and start a new sentence stating AAP recommendations

Line 74 consider rephrasing “Better and …” such as “In an effort to improve and standardized NAS treatment, there is a need for xxx.”

AUTHORS’ RESPONSE: We have made all edits to the introduction as suggested by the reviewer. 

Page 10

Line 123 consider replacing newborn infants with neonates

Line 157 delete “very small”

AUTHORS’ RESPONSE: We have replaced “newborn infants” with “neonates” throughout, per suggestions from both reviewers.

Page 11 Results and Discussion consider separating these two sections out

AUTHORS’ RESPONSE: We have separated out the results and discussion sections based on comments from both reviewers.

Line 164 indent, consider rephrasing this sentence such as “The incidence of NAS in Arizona continues to rise..”

Line 166 consider rephrasing your sentence on “the large number of infants… “ recommend separating this point from the sentence and highlight the incidence of NAS is a sequela/consequence from the opioid epidemic

AUTHORS’ RESPONSE: We have made both of the suggested revisions noted above.

172 Indent, again consider neonate rather than “newborn infant” and rephrase this sentence for clarity.

AUTHORS’ RESPONSE: We have made this change throughout.

181 Indent. Curious are these rates including infants treated in the intensive care, meaning are these NAS neonates of all gestational ages compared to all conditions/neonates. Would be a great point to highlight if so..

AUTHORS’ RESPONSE: Yes, these rates include infants treated in intensive care. We have added text to better point that out (methods section, paragraph 2).

191 Indent.

205 Rephrase “past studies,” consider “Previous studies reported…”

AUTHORS’ RESPONSE: We have made the suggested change.

Page 12

207 delete duplicate that

232 consider rephrasing “we are planning..” consider “future studies may investigate..”

AUTHORS’ RESPONSE: We have made the changes suggested above.

Page 13

264 Discussion/Conclusions – Consider deleting discussion

AUTHORS’ RESPONSE: We have separated out the discussion and conclusion sections as recommended by both reviewers.

---

## [Decision Letter · Decision Letter 1]

1 Mar 2021

Characterization of Neonatal Abstinence Syndrome in Arizona from 2010-2017

PONE-D-20-17306R1

Dear Dr. Hepp,

We’re pleased to inform you that your manuscript has been judged scientifically suitable for publication and will be formally accepted for publication once it meets all outstanding technical requirements.

Kind regards,

Barbara Wilson Engelhardt, MD

Academic Editor

PLOS ONE

Additional Editor Comments (optional):

Dear Dr. Hepp,

In their 2nd review of your article, following your revision, both reviewers found adequate response to their comments/concerns. They accepted the revised paper.

I concur and support accepting your paper. Your epidemiologic work and results are interesting and contributing significantly to the understanding of the current Opioid epidemic.

I have 2 minor concerns:

First, you make mention of the work by Dr. Stephen Patrick's group several times throughout your paper. It is not necessary to use his name in the abstract.

Secondly you use the term NAS and may want to consider the now often used NOWS in your title and as the predominant term throughout the paper.

All the best,

B Engelhardt, Academic Editor

Reviewers' comments:

Reviewer's Responses to Questions

**Comments to the Author**

1. If the authors have adequately addressed your comments raised in a previous round of review and you feel that this manuscript is now acceptable for publication, you may indicate that here to bypass the “Comments to the Author” section, enter your conflict of interest statement in the “Confidential to Editor” section, and submit your "Accept" recommendation.

Reviewer #1: All comments have been addressed

Reviewer #2: All comments have been addressed

2. Is the manuscript technically sound, and do the data support the conclusions?

Reviewer #1: Yes

Reviewer #2: Yes

3. Has the statistical analysis been performed appropriately and rigorously? 

Reviewer #1: Yes

Reviewer #2: Yes

4. Have the authors made all data underlying the findings in their manuscript fully available?

Reviewer #1: Yes

Reviewer #2: Yes

5. Is the manuscript presented in an intelligible fashion and written in standard English?

Reviewer #1: Yes

Reviewer #2: Yes

6. Review Comments to the Author

Reviewer #1: I appreciate the authors' attention to their revisions. I have no further comments or concerns about the publication of this submission. Excellent work!

Reviewer #2: (No Response)

7. PLOS authors have the option to publish the peer review history of their article (what does this mean?). If published, this will include your full peer review and any attached files.

Reviewer #1: **Yes: **Laurie L. Meschke

Reviewer #2: No

---

## [Editor Report · Acceptance letter]

28 Apr 2021

PONE-D-20-17306R1 

Characterization of Neonatal Opioid Withdrawal Syndrome in Arizona from 2010-2017 

Dear Dr. Hepp:

I'm pleased to inform you that your manuscript has been deemed suitable for publication in PLOS ONE. Congratulations! Your manuscript is now with our production department. 

Kind regards, 

on behalf of

Dr. Barbara Wilson Engelhardt 

Academic Editor

PLOS ONE